# Imaging of Peritoneal Metastases in Ovarian Cancer Using MDCT, MRI, and FDG PET/CT: A Systematic Review and Meta-Analysis

**DOI:** 10.3390/cancers16081467

**Published:** 2024-04-11

**Authors:** Athina C. Tsili, George Alexiou, Martha Tzoumpa, Timoleon Siempis, Maria I. Argyropoulou

**Affiliations:** 1Department of Clinical Radiology, Faculty of Medicine, School of Health Sciences, University of Ioannina, University Campus, 45110 Ioannina, Greece; martz_me@hotmail.com (M.T.); margyrop@uoi.gr (M.I.A.); 2Department of Neurosurgery, Faculty of Medicine, School of Health Sciences, University of Ioannina, University Campus, 45110 Ioannina, Greece; galexiou@uoi.gr; 3ENT Department, Ulster Hospital, Upper Newtownards Rd., Dundonald, Belfast BT16 1RH, UK; timoleon.siempis@setrust.hscni.net

**Keywords:** peritoneal carcinomatosis, ovarian cancer, computed tomography, multidetector, magnetic resonance imaging, diffusion weighted MRI, PET/CT scan

## Abstract

**Simple Summary:**

Ovarian cancer is the leading cause of death due to gynecologic malignancies. Peritoneal metastases represent the most common pathway for the spread of OC, both at the time of initial diagnosis and at recurrence. Accurate mapping of peritoneal metastases helps in planning the appropriate therapeutic strategy, predicting the likelihood of optimal cytoreduction, and identifying potentially unresectable or difficult disease sites that may require surgical technique modifications. Preoperative diagnostic work-up with multidetector CT (MDCT), MRI, including diffusion-weighted imaging (DWI), or FDG PET/CT plays a vital role in the accurate assessment of the extent of peritoneal carcinomatosis. In this article, the aim was to update the role of MDCT, MRI, including DWI, and FDG PET/CT in the detection of peritoneal metastases in ovarian cancer by conducting a systematic review and meta-analysis of the existing literature.

**Abstract:**

This review aims to compare the diagnostic performance of multidetector CT (MDCT), MRI, including diffusion-weighted imaging, and FDG PET/CT in the detection of peritoneal metastases (PMs) in ovarian cancer (OC). A comprehensive search was performed for articles published from 2000 to February 2023. The inclusion criteria were the following: diagnosis/suspicion of PMs in patients with ovarian/fallopian/primary peritoneal cancer; initial staging or suspicion of recurrence; MDCT, MRI and/or FDG PET/CT performed for the detection of PMs; population of at least 10 patients; surgical results, histopathologic analysis, and/or radiologic follow-up, used as reference standard; and per-patient and per-region data and data for calculating sensitivity and specificity reported. In total, 33 studies were assessed, including 487 women with OC and PMs. On a per-patient basis, MRI (*p* = 0.03) and FDG PET/CT (*p* < 0.01) had higher sensitivity compared to MDCT. MRI and PET/CT had comparable sensitivities (*p* = 0.84). On a per-lesion analysis, no differences in sensitivity estimates were noted between MDCT and MRI (*p* = 0.25), MDCT and FDG PET/CT (*p* = 0.68), and MRI and FDG PET/CT (*p* = 0.35). Based on our results, FDG PET/CT and MRI are the preferred imaging modalities for the detection of PMs in OC. However, the value of FDG PET/CT and MRI compared to MDCT needs to be determined. Future research to address the limitations of the existing studies and the need for standardization and to explore the cost-effectiveness of the three imaging modalities is required.

## 1. Introduction

Ovarian cancer (OC) represents the fifth most commonly diagnosed cancer among women, the fifth cause of cancer death, and the commonest cause of death due to gynecologic malignancies [1,2,3,4,5,6,7,8]. An estimated number of 19,680 new cases of OC are expected to be diagnosed in the US and 12,740 women are expected to die from the disease in 2024 [2]. Most cases (90%) are epithelial ovarian carcinomas and the majority are high-grade serous carcinomas. The most important revision in the last FIGO staging classification is that ovarian, fallopian, and primary peritoneal cancers are considered as one entity [9].

Ovarian cancer has a poor prognosis, with a 5-year relative survival rate of 48%, mainly because most women are diagnosed with advanced-stage disease [2]. Moreover, the percentage of recurrence in OC is very high. The standard of care in OC includes either primary debulking surgery (PDS) followed by adjuvant chemotherapy or neoadjuvant chemotherapy (NAC) prior to interval debulking surgery (IDS) and postoperative chemotherapy [10,11,12].

Peritoneal metastases (PMs) represent the commonest pathway for the spread of OC and are often seen either at the time of initial diagnosis or at recurrence [3,13,14,15,16,17,18,19,20,21,22]. The peritoneal cancer index (PCI) introduced by Jacquet and Sugarbaker combined the distribution of PMs in 13 abdominopelvic regions (ARs) with the tumor size providing a measurement of the volume of peritoneal carcinomatosis (PC) and also a valuable prognostic index (Table 1, Figure 1) [23].

Imaging has a fundamental role in the accurate diagnosis of PMs in OC, helping to plan the appropriate therapeutic strategy, predict the likelihood of optimal cytoreduction, and identify potentially unresectable or difficult disease sites, which may require either IDS following chemotherapy or surgical technique modifications during PDS [3,13,14,15,16,17,18,19,20,21,22].

Multidetector CT (MDCT) is considered the examination of choice for the initial staging of OC and for the evaluation of the extent of the disease in suspected recurrence [1]. However, CT has limitations, mainly low soft-tissue resolution, and difficulty in depicting small peritoneal implants or implants at certain anatomic areas, including the root of mesentery, lesser omentum, and serosal surfaces of the small bowel, especially in the absence of ascites [1,3,18,21,24,25,26,27,28,29,30].

MRI represents another reliable imaging tool for the assessment of PC. The efficacy of the technique in the detection of PMs has been improved by using fat-suppressed delayed contrast-enhanced imaging and diffusion-weighted imaging (DWI) [16,21,30,31,32,33,34,35]. Specifically, DWI improves the detection of small peritoneal hypercellular implants, even in the absence of ascites, due to their high signal against the hypointense background of normal tissues. However, MRI is recommended in specific circumstances, such as women with borderline ovarian tumors or OCs that have been previously staged with fertility preservation and also in cases with inconclusive CT findings [1].

The hypermetabolic activity of PMs increases their conspicuity using FDG PET/CT. CT and FDG PET/CT are considered equivalent alternatives for the detection of recurrent OC [1,17,30,36,37,38,39,40,41,42,43,44]. Based on the results of a recently published meta-analysis, FDG PET/CT had high diagnostic accuracy, with 88% sensitivity and 89% specificity in the detection of recurrent OC [45]. Up to now, FDG PET or FDG PET/CT may be used as an adjunct tool in the initial staging of OC, in cases of indeterminate CT findings [1].

Systematic reviews on the role of cross-sectional imaging in the detection of PC in women with OC are lacking. A few recently published meta-analyses assessed the diagnostic accuracy of imaging modalities in the detection of PMs from various primary malignancies, including OC [46,47,48].

The purpose of this systematic review and meta-analysis was to compare the diagnostic performance of MDCT, MRI, including DWI, and FDG PET/CT in the detection of peritoneal metastases in ovarian cancer.

## 2. Materials and Methods

This systematic review and meta-analysis was conducted in accordance with the preferred reporting items for systematic reviews and meta-analysis (PRISMA) guidelines [49]. The systematic review has not been registered.

### 2.1. Search Strategy

A systematic and comprehensive literature search was performed for all publications that reported the diagnostic performance of MDCT, MRI, and FDG PET/CT in the detection of PMs in OC. Data extraction was independently performed by two researchers (ACT and MT) from the PubMed/MEDLINE database and included articles published from 2000 to February 2023.

The following keywords were used: “ovarian cancer” OR “peritoneal metastases” OR “peritoneal carcinomatosis” OR “multidetector CT” OR “MDCT” OR “magnetic resonance imaging” OR “MRI” OR “diffusion-weighted imaging” OR “DWI” OR “fluorine-18-fluorodeoxyglucose (FDG) positron emission tomography (PET)/computed tomography (CT)” and “FDG PET/CT”.

Articles found to be suitable on the basis of their title and abstract were subsequently selected to further determine appropriateness for inclusion in this meta-analysis. Only papers in the English language were assessed. Full-text studies were further evaluated, and exclusion criteria were applied to identify final papers for inclusion. References were manually screened to identify additional studies.

### 2.2. Eligibility Criteria

The inclusion criteria were as follows: diagnosis/suspicion of PMs in patients with ovarian/fallopian/primary peritoneal cancer; initial staging or suspicion of recurrence (primary outcome); MDCT, MRI, and/or FDG PET/CT performed for the detection of PMs; population of at least 10 patients; surgical results, histopathologic analysis, and/or radiologic follow-up, used as a reference standard; per-patient and per-region data included; and data for calculating sensitivity and specificity reported. Discrepancies regarding potential eligibility and inclusion were resolved by consensus.

Studies were excluded if results for different imaging modalities were presented in combination and if data on the performance of each individual technique were unavailable. Studies including patients with the diagnosis of PMs from tumors other than OC were considered eligible only if it was possible to extrapolate results obtained on PC from OC.

### 2.3. Data Extraction

From each study, the following design characteristics were recorded: first author and year of publication; study design (prospective or retrospective); primary outcome; characteristics of study population, including number of patients with ovarian/fallopian/primary peritoneal cancer, age, number of patients with PMs, number and size of PMs, location of PMs; imaging modality, including MDCT, MRI or FDG PET/CT; report of reference test; time interval between imaging modalities; and time interval between imaging and reference standard.

Imaging characteristics included detailed information on the following: imaging equipment (type of scanner for MDCT and FDG PET/CT, magnetic field strength); imaging technique (phases and reformations for MDCT, type of coil, sequences, section thickness, and *b*-values for MRI); bowel preparation (laxatives and spasmolytic drugs), and use of luminal and/or intravenous contrast medium.

The numbers of true-positive (TP), false-negative (FN), false-positive (FP), and true-negative (TN) results for the detection of PMs were extracted on a per-patient and per-region basis. When cumulative data on the detection of PMs were not reported, the results from the abdominopelvic region with the best diagnostic performance were included in the analysis.

Regarding the per-patient analysis, the diagnostic odds ratios (DORs) were also estimated. A bivariate random effect meta-analytic method was used to estimate pooled sensitivity, specificity, and summary receiver operating characteristic (SROC) curves. Via the percentage of heterogeneity between the studies, computing I^2^ values were calculated. I^2^ values equal to 25%, 50%, and 75% were assumed to represent low, moderate, and high heterogeneity, respectively. Study heterogeneity was also assessed visually via funnel plots.

When at least three datasets were available for the three imaging modalities, subgroup analyses were performed for the different ARs, including AR0 (central abdomen), AR1 (right hypochondrium), AR2 (epigastrium), AR3 (left hypochondrium), AR4 (left lumbar region), AR5–7 and AR6 (pelvis), AR8 (right lumbar region), small bowel (AR9–12), colon and mesentery (Table 1, Figure 1), on a per-patient and a per-region basis [23].

### 2.4. Quality Assessment

Analyses were performed using the RevMan software (ReviewManager, version 5.3; The Nordic Cochrane Centre, The Cochrane Collaboration, Copenhagen, Denmark). To assess the methodological quality of the included primary studies and to detect potential bias, we used the quality assessment of diagnostic accuracy studies-2 (QUADAS-2) tool [50].

## 3. Results

The initial search in the electronic database resulted in 848 articles. Following a review of the titles and abstracts, 187 studies were selected as potentially relevant, and their references were cross-checked. Thirty-three publications eventually fulfilled the inclusion criteria and were selected for quantitative synthesis (Table 2) [26,27,30,42,44,51,52,53,54,55,56,57,58,59,60,61,62,63,64,65,66,67,68,69,70,71,72,73,74,75,76,77,78]. The flow chart of the selection process is shown in Figure 2.

A total of 2025 women with OC were included in the meta-analysis, with a mean age of 57.4 years (range, 19–91 years). The primary outcome included 19 studies with initial diagnosis of ovarian/fallopian/peritoneal cancer, 7 reports with recurrent cancer, and 7 studies with primary or recurrent OC. Advanced OC (FIGO stages III and IV) was reported in 1453 patients (Table 2). The presence of PMs was reported in 487 women and included 4.588 ARs (Table 2 and Table 3).

### 3.1. Study Characteristics

The studies selected for meta-analysis included 15 prospective and 18 retrospective articles (Table 2). In total, 23 datasets evaluated the presence of PMs in OC with MDCT [26,27,30,52,56,57,58,59,63,64,65,66,67,68,69,70,71,72,73,74,75,76,78] (Table 4); 2, 3, and 8 studies were performed on a 4-row [52,58], 16-row [56,57,74], and 64-row [27,30,59,63,67,70,72,75] MDCT scanner, respectively, 4 studies used both a 16-row and a 64-row CT [64,66,73,76], and 1 report used a 4-row, a 16-row, and a 64-row [65] CT machine (Table 5). Overall, 20 studies [27,30,52,56,57,58,59,63,64,65,66,67,68,69,70,71,72,73,74,76,78] reported the intravenous administration of iodinated contrast medium; 10 of them used the portal phase [27,56,57,58,59,66,67,68,73,78], 4 used both the arterial and the portal phase [52,65,74,76], 1 used the arterial phase [72], and 1 study used both the portal and the delayed phase [69]. The use of luminal contrast was reported in 15 studies [30,52,64,65,66,67,68,69,70,71,72,73,74,76,78]; 8 studies reported the oral administration of H_2_O and diluted contrast medium, including iodinated contrast material in 4 reports [30,66,74,78], gastrografin in 2 studies [69,72], mannitol in 1 study [70], Macrogol in 1 report [65], and 2 studies used H_2_O, administered orally in 1 report [52] and as a rectal enema in 1 study [67]. Multiplanar reformations (MPRs) used for data interpretation were reported in 11 studies [27,52,56,57,59,65,67,69,70,73,78]; the application of coronal and sagittal reformations was reported in 4 studies [27,56,57,78]; coronal, sagittal, and oblique MPRs were used in 2 studies [52,65]; coronal plane in 2 reports [59,73]; and in 1 report [70] combined MPRs with three-dimensional maximum intensity projection reformations were performed (Table 5).

MRI was used in 10 studies [30,51,53,55,61,62,66,67,73,78], 5 performed on a 1.5 T [30,51,53,55,61] and 5 on a 3.0 T system [62,66,67,73,78] (Table 4 and Table 6). DWI was applied in six studies [30,62,66,67,73,78], including three reports with whole-body DWI (WB-DWI) [66,73,78] (Table 6). Gadolinium chelate was administered intravenously in nine studies [51,53,55,61,62,66,67,73,78]; two studies reported the use of the portal phase [66,67] and one study used three post-contrast phases (arterial, portal, and delayed) [61]. The administration of luminal contrast prior to the MRI was reported in four studies [66,67,73,78]; two of them reported the use of pineapple juice [66,73], one study used both H_2_O and pineapple juice [78], and, in one study, H_2_O was given via the rectum [67]. Bowel preparation with intravenous or intramuscular administration of spasmolytic agents was reported in eight studies [30,51,53,55,66,67,73,78] (Table 6).

FDG PET/CT was used in 16 studies [30,42,44,54,55,56,57,60,61,63,64,66,67,68,70,77], including 7 reports with diagnostic contrast-enhanced CT (CECT) [30,56,57,63,66,70,77] (Table 4 and Table 7). Six studies were performed on a 16-row [42,56,57,64,67,68] scanner, four on a 64-row [30,63,70,77] system, and three studies, each one used a 4-row [54], an 8-row [55], and a spiral [66] CT machine (Table 7).

The following reference tests were used: surgical and histopathologic results (*n* = 21) [26,30,42,44,51,53,54,57,60,62,63,64,65,67,71,72,74,75,76,77,78], surgical findings (*n* = 3) [52,58,68], surgical and histopathologic results or follow-up (*n* = 8) [27,55,56,59,61,66,70,73], and follow-up (*n* = 1) [69] (Table 4).

### 3.2. Quality Assessment

Study quality grading using QUADAS II scores showed that in terms of risk bias and regarding patient selection, index test, reference standard, flow, and timing, the majority of studies included in the analysis were of high to good quality (Figure 3). In terms of applicability, the quality of reporting on patient selection, index test, and the gold standard was good (Figure 3).

### 3.3. Diagnostic Performance

#### 3.3.1. Per-Patient Analysis

In total, 15, 7, and 8 datasets for MDCT [26,27,30,52,63,64,65,69,70,71,72,73,74,75,78], MRI, including DWI [30,51,53,61,62,73,78], and FDG PET/CT [30,44,54,61,63,64,70,77], respectively, were included in the per-patient analysis. The sensitivity estimates for MDCT, MRI, and FDG PET/CT on a per-patient basis were 79.7% (95% confidence interval [CI], 75.6–83.4%, I^2^ = 82.6%), 82.7% (95% CI, 76.0–88.2%, I^2^ = 87.9%), and 93.7% (95% CI, 90.0–96.3%, I^2^ = 20.4%), respectively. The specificity estimates for MDCT, MRI, and FDG PET/CT on a per-patient basis were 92.1% (95% CI, 89.6–94.2%, I^2^ = 85.1%), 90.3% (95% CI, 86.7–93.1%, I^2^ = 86.4%), and 91.5% (95% CI, 86.8–95.0%, I^2^ = 84.6%), respectively (Figure 4 and Figure 5). Figure 6 shows study heterogeneity via funnel plots.

The per-patient diagnostic performance, including TP, FN, FP, and TN findings; sensitivity; specificity; positive predictive value (PPV); and negative predictive value (NPV) for each imaging modality are presented in Appendix A. The DOR estimates for MDCT, MRI, and FDG PET/CT on a per-patient basis were 29.55% (95% CI, 17.54–49.78%), 93.95% (95%CI, 27.41–321.97%), and 84.15% (95%CI, 17.62–401.8%), respectively. The summary area-under-the-curve (SAUC) for MDCT, MRI, and FDG PET/CT was determined to be 0.91, 0.96, and 0.97, respectively (Figure 7). The sensitivity estimates for MRI (*p* = 0.03) and FDG PET/CT (*p* < 0.01) were higher than that for MDCT, on a per-patient basis. FDG PET/CT had higher sensitivity compared to MRI, although non-significant (*p* = 0.84).

#### 3.3.2. Per-Region Analysis

Per-region data were analyzed for 338 women with advanced OC and 3.881 ARs (Appendix A). Overall, 12, 3 and 11 datasets for MDCT [27,56,57,58,59,63,65,66,67,68,69,76], MRI, including DWI [55,66,67], and FDG PET/CT [42,54,55,56,57,60,63,66,67,68,77], respectively, were included in the per-region analysis. On a per-region basis, comparison between MDCT, MRI, and FDG PET/CT for detecting PMs revealed a sensitivity of 70.1% (95% CI, 68.5–71.6% I^2^ = 98.9%), 92.6% (95% CI, 89.0–95.3%, I^2^ = 73.2%), and 58.3% (95% CI, 56–60.6%, I^2^ = 96.8%), respectively. The specificity estimates for MDCT, MRI, and FDG PET/CT were 90.2% (95% CI, 89.3–91.1%, I^2^ = 97.4%), 90.3% (95% CI, 86.7–93.2%, I^2^ = 72.6%), and 92.6% (95% CI, 91.4–93.7%, I^2^ = 92.6%), respectively (Figure 8 and Figure 9). The per-region diagnostic performances are presented in Appendix A. MDCT, MRI, and FDG PET/CT had an AUC of 0.92, 0.96, and 0.89, respectively, on a per-region analysis (Figure 10). No differences in sensitivity estimates were found between MDCT and MRI (*p* = 0.25), MDCT and FDG PET/CT (*p* = 0.68), and MRI and FDG PET/CT (*p* = 0.35), on a per-region basis.

#### 3.3.3. Subgroup Analysis: Abdominopelvic Regions

##### Per-Patient Analysis

Table 8, Table 9 and Table 10 show sensitivity and specificity estimates for the three imaging modalities in different ARs, on a per-patient basis.

Datasets for assessing per-patient diagnostic accuracy of MDCT were available for all ARs, including AR0 [52,63,71,72,74,75,78], AR1 [26,30,52,63,72,74,75,78], AR2 [52,72,74,78], AR3 [26,52,63,72,74,75,78], AR4 [72,74,78], AR5–7 [26,52,63,71,72,74,75,78], AR6 [26,52,71,72,74,75,78], AR8 [72,74,78], diaphragm [26,52,63,71,72,78], small bowel [26,63,65,72,74,75,78], colon [26,63,72,78], and mesentery [26,30,52,63,71,72,73,75,78] (Appendix A). The accuracy of MDCT in the detection of PMs on a per-patient basis was higher in six ARs: the left hypochondrium (AR3), including the undersurface of the left hemidiaphragm, spleen, pancreatic tail of the pancreas, and anterior and posterior surfaces of the stomach, with sensitivity estimates of 61.8% (95% CI, 50.9–71.9%), specificity estimates 97.9% (95% CI, 95.8–99.2%), and an AUC of 0.93; the diaphragm, with a sensitivity of 49.7% (95% CI, 42.6–56.9%), a specificity of 97.7% (95% CI, 94.8–99.3%), and an AUC of 0.91; the pelvis–hypogastrium (AR6), including the female internal genitalia, the urinary bladder, the cul-de-sac of Douglas, and the rectosigmoid colon, with a sensitivity of 66.2% (95% CI, 59.7–72.3%), a specificity of 93.3% (95% CI, 89.7–95.9%), and an AUC of 0.92; and the central abdomen (AR0), including the midline abdominal incision, the greater omentum, and the transverse colon, with sensitivity estimates of 80.1% (95% CI, 74.5–84.9%), specificity estimates 89% (95% CI, 83.1–93.3%), and an AUC of 0.91; left lumbar region (AR4), including the descending colon and the left paracolic gutter, with sensitivity estimates of 73% (95% CI, 60.3–83.4%), specificity estimates 86.3% (95% CI, 76.7–92.9%), and an AUC of 0.92; and pelvis (A5–7), with a sensitivity of 64.1% (95% CI, 58.4–69.4%), a specificity of 95.1% (95% CI, 91.6–97.4%), and an AUC of 0.90 (Table 8). MDCT had the lowest diagnostic performance on a per-patient basis in two regions: the colon, with a sensitivity of 30.5% (95% CI, 23.2–38.5%), a specificity of 95.8% (95% CI, 92.2–98.1%), and an AUC of 0.36, and the mesentery, with a sensitivity of 33.8% (95% CI, 27.2–41%), a specificity of 96.9% (95% CI, 94.9–98.3%), and an AUC of 0.66 (Table 8).

Datasets for assessing the per-patient diagnostic accuracy of MRI were available for only three ARs, including, AR0 [51,53,78], the diaphragm [51,53,62,78], and the mesentery [30,51,62,73,78] (Appendix A). The sensitivity (59.2%) and specificity (75.7%) of MRI were higher in the mesentery, with an AUC of 0.90 (Table 9).

Available data for assessing the diagnostic performance of FDG PET/CT on a per-patient basis included five ARs: AR0 [44,63,77], AR1 [30,44,63,77], AR3 [44,63,77], A5–7 [44,63,77], and mesentery [30,63,77] (Appendix A). Detection rates for FDG PET/CT were higher in two regions: the central abdomen (AR0), with sensitivity estimates of 92.9% (95% CI, 86.5–96.9%), specificity estimates of 85.2% (95% CI, 73.8–93%), and an AUC of 0.95, and the pelvis (AR5–7), with sensitivity estimates of 91.5% (95% CI, 85–95.9%), specificity estimates of 87.5% (95% CI, 74.8–95.3%), and an AUC of 0.94 (Table 10).

##### Per-Region Analysis

On a per-region analysis, data assessing the diagnostic accuracy were available for MDCT in four regions, including AR0, AR5–7, and AR6, the diaphragm and mesentery [27,57,58,66], and for FDG PET/CT in six regions, including AR0 [42,57,60,66], AR1 [42,54,60,66], AR3 [42,54,60,66], AR4 [42,60,66], AR8 [42,60,66], and the pelvis (AR5–7) [42,54,57,60,66] (Appendix A). The assessment of MRI data in different ARs on a per-region basis was not possible due to the small number of studies.

The highest detection rates for MDCT on a per-region analysis were noted at the pelvis–hypogastrium [sensitivity, 24.4% (95% CI, 12.4–40.3%); specificity, 96.4% (95% CI, 89.9–99.3%); and, AUC, 0.99] and the lowest detection rates were found at the mesentery [sensitivity: 43.4% (95% CI, 29.8–57.7%); specificity, 90.7% (95% CI, 83.6–95.5%); and, AUC: 0.68] (Table 11). The sensitivity for detecting PMs in different ARs for FDG PET/CT on a per-region basis revealed better results in the right hypochondrium [sensitivity: 66.7% (95% CI, 54.8–77.1%); specificity, 81.8% (95% CI, 69.1–90.9%); and, AUC: 0.99] (Table 12).

## 4. Discussion

According to our knowledge, this is an up-to-date systematic review and meta-analysis that exclusively compares the diagnostic performance of MDCT, MRI, including DWI, and FDG PET/CT in the detection of peritoneal metastases in women with ovarian cancer. In total, 33 studies, 23 using MDCT, 10 MRI (including three reports with DWI and three studies with WB-DWI), and 16 using FDG PET/CT (including seven studies with CECT), were evaluated. On a per-patient basis, FDG PET/CT had the highest sensitivity (93.7%) when compared to MRI (82.7%) and MDCT (79.7%). Specificity estimates were high for all imaging modalities (92.1%, 90.3%, and 91.5% for MDCT, MRI, and FDG PET/CT, respectively). Both FDG PET/CT and MRI have comparably higher per-patient diagnostic accuracy for the detection of PMs when compared to MDCT.

No differences in the diagnostic performance between MDCT, MRI, and FDG PET/CT were found on a per-lesion basis. MRI had the highest sensitivity (92.6%), when compared to MDCT (70.1%) and FDG PET/CT (58.3%), although our results are limited, due to the small number of MRI datasets (*n* = 3). Specificity estimates were comparably high for all imaging modalities (90.2%, 90.3%, and 92.6% for MDCT, MRI, and FDG PET/CT, respectively). Based on the results of this meta-analysis, FDG PET/CT and MRI had higher sensitivity compared to MDCT in the detection of PMs in OC.

Similar to our results, a recently published meta-analysis reported comparable diagnostic performance for DWI MRI and FDG PET/CT, higher than that of CT for the detection of PMs in ovarian and gastrointestinal cancer patients [47]. This review was based on 28 articles, including 20, 7, and 10 CT, DWI MRI, and FDG PET/CT datasets, respectively. The pooled sensitivity and specificity were 68% and 88% for CT, 92% and 85% for DWI MRI, and 80% and 90% for FDG PET/CT [47].

MDCT is routinely used for the preoperative imaging of primary OC, with a reported staging accuracy of up to 94% [1,3,5,6,18,19,20,21,69,79]. Portal venous phase and water density oral contrast usually provide detailed mapping of PMs [1,69]. MDCT is also used for the evaluation of any persistence of disease after CRS and during follow-up, with a sensitivity and specificity of 58–84% and 59–100%, respectively, in the detection of OC recurrence [1,69,70,80].

The main advantages of MDCT include the following: wide availability, rapid scanning, increased volume coverage, excellent spatial resolution, robustness, and reproducibility of image acquisition. CT is devoid of misregistration artifacts and allows the acquisition of thin sections and the creation of high-resolution MPRs, improving the detection of small PMs, especially when a large amount of ascites is present, and the detailed exploration of curved peritoneal surfaces [18,19,20,21,67,69,70,81,82,83,84,85]. Coronal reformations improve the assessment of hemidiaphragms, hepatic and splenic surfaces, and paracolic gutters and the evaluation of the extent of omental disease. Sagittal MPRs improve the assessment of the hemidiaphragms, the Douglas pouch, the vaginal cuff, the peritoneal surface of the bladder, and the rectosigmoid colon [83,84,85]. The use of multiplanar reformations was reported in 11 articles in this meta-analysis, although comparative studies on the diagnostic performance of axial images versus MPRs in the detection of PMs were not performed due to limited data [27,52,56,57,59,65,67,69,70,73,78].

However, CT has limitations, including poor soft tissue contrast and reduced sensitivity for the detection of small PMs (<5 mm) and those in certain anatomical locations (e.g., mesentery and bowel serosa), especially in the absence of ascites [3,13,18,19,20,21,25,26,66,67,69,70,73,81,82]. Subgroup analysis including PMs of different sizes was not performed in the present review, due to inadequate relevant data.

MDCT comprised the largest dataset in our meta-analysis, allowing a comprehensive assessment of the diagnostic accuracy of the technique in the detection of PMs in different abdominopelvic regions. The highest MDCT detection rates were noted at the left hypochondrium (AR3), the central abdomen (AR0), the diaphragm, the pelvis (AR5–7 and AR6), and the left lumbar region (AR4). Our observations are primarily related to the advantages of MDCT technology, namely, the acquisition of thin slices and the creation of high-resolution reformations, resulting in an improvement in the evaluation of curved structures, such as the undersurface of the diaphragms, the paracolic gutter, and the pelvis [3,19]. Similar to published data, this review confirmed the low diagnostic performance of CT in the colon and the mesentery [3,13,20,25,26,73,81]. The detection of early mesenteric involvement or small-sized serosal bowel PMs may be problematic, as CT signs may be subtle, especially in the absence of adequate bowel distention [3].

Based on the analysis of 10 datasets, MRI proved more accurate compared to MDCT for the detection of PMs, on a per-patient basis. The use of fat suppression, delayed contrast-enhanced sequences, and DWI contribute to the improvement in the accuracy of MRI in the detection of PMs [3,5,16,20,21,30,31,32,33,34,35,66,69,86,87,88,89,90,91]. MRI allows better detection of subcentimeter PMs and PC involving certain anatomic areas, such as the bowel serosal surface, the pelvis, the right hypochondrium, and the mesentery. The interobserver agreement of MRI has also been reported to be higher compared to CT in most ARs [31,81]. Although the diagnostic performance of MRI was only assessed in three ARs, including the central abdomen (AR0), the diaphragm, and the mesentery, our systematic review found that MRI was more accurate in the bowel mesentery.

Normal peritoneal enhancement is equal to or less than that of the liver. Contrast enhancement greater than the liver is abnormal and may represent the only finding suggestive of PC. This sign is not always detected by MDCT; however, it is readily appreciated by MRI on delayed fat-suppressed contrast-enhanced imaging [3,5,20,81,88]. The sensitivity of MRI for PMs has been reported to increase by using DWI in combination with conventional MRI sequences, even in the absence of ascites. The increased contrast between the hyperintense hypercellular implants against the surrounding hypointense normal tissues enhances the detectability of PC by DWI [5,20,21,31,81,86,87,88,89].

No direct comparison between the accuracy of conventional MRI sequences and DWI was performed in this analysis, due to limited data.

Limitations of MRI are related to the high cost and long examination time, motion artifacts, lack of routine use of intraluminal contrast agents, and need for experience in image acquisition and interpretation [1,20,81,92]. MRI is also limited by its ability to detect small, calcified PMs, which are easily detected by MDCT. Disadvantages related to DWI are due to the low spatial resolution; presence of false-positives, attributed to densely cellular tissue, such as fibrosis, bowel mucosa, endometrium, and abscess; and false-negatives, attributed to mucinous carcinomas and well-differentiated malignancies [1,20].

Similar to MRI, this meta-analysis showed that FDG PET/CT was more sensitive than MDCT in the assessment of PC in OC on a per-patient basis. The main advantage of the technique is the whole-body coverage. FDG PET/CT can detect small PMs; evaluate all peritoneal compartments, even those inaccessible during surgery, such as the subdiaphragmatic peritoneal surfaces and the bowel mesentery; better assess ascites; and discriminate nodular peritoneal implants from the intestinal loops [1,5,17,30,36,37,38,39,40,41,42,43,44,63,67,68,69,93,94,95,96,97,98,99,100,101]. The present review showed that FDG PET/CT had the highest detection rates in the central abdomen (AR0), the right hypochondrium (AR1), and the pelvis (AR5–7).

FDG PET/CT disadvantages are related to limited spatial resolution in the detection of small PMs (<5 mm), difficulty in the evaluation of diffuse peritoneal disease, presence of tissues with low FDG avidity, such as mucinous tumors, and possible discrepancy in lesion location between CT and PET/CT caused by respiratory movements and intestinal peristalsis. False positives may be due to inflammation, infection, and benign conditions or the normal physiological activity in the bowel, gallbladder, vessels, ureters, and urinary bladder. Shortcomings of PET/CT also include the limited availability and the high cost [5,67,68,101].

Complete resection of all macroscopic peritoneal implants has been proven to be the single most important independent prognostic factor in OC. Diagnostic laparoscopy can provide a definitive histologic diagnosis and detailed information on the extent of PC. However, up to 40% of women may be understaged surgically as small PMs in areas such as the subdiaphragmatic surfaces, the porta hepatis, or the hepatorenal fossa are not easily accessible. In addition, diagnostic laparoscopy has been associated with a high incidence of port-site metastases, although these do not worsen the patient’s prognosis [1,79]. Preoperative diagnostic work-up with CT, MRI, or FDG PET/CT is vital in the assessment of the extent of PC in OC [1,79].

Tumor heterogeneity in OC at a cellular and genetic level is a well-known phenomenon that cannot be thoroughly evaluated using conventional imaging data. Quantitative semi-automated and automated methods based on artificial intelligence techniques have been developed, which can be applied to routine medical images to assess tumor heterogeneity. The use of radiomics and radiogenomics may be helpful in the future in predicting OC genotype and biology and in assessing treatment response, clinical outcome, and patient survival [102,103,104,105,106]. Based on preliminary data, MRI and CT-based radiomics have been reported to predict the presence of PMs in OC [107,108,109,110].

This meta-analysis has inherent limitations, mainly related to publication bias and study heterogeneity. Our systematic review was limited to the PubMed database, including published studies reporting a “positive effect” that might overestimate the actual magnitude of an effect. However, study quality grading showed that most of the studies included in the analysis were of high to good quality.

Heterogeneity among included patient groups is another shortcoming, due to differences in primary outcome (primary staging and recurrent disease). Subgroup analysis assessing the differences in the diagnostic performance of MDCT, MRI, and FDG PET/CT in the detection of PMs between primary and recurrent OC was not performed, due to the lack of relevant data. Heterogeneity in study design, imaging methodologies (including scanners, protocols, sequences, and intravenous/oral contrast), reader experience, and reference standards (ranging from histopathologic confirmation to surgical findings and imaging follow-up) is another limitation. The standardization of imaging techniques and consensus on the interpretation criteria for PMs across different centers would facilitate more accurate and reliable assessments. Finally, no data on the cost-effectiveness of MDCT, MRI, and FDG PET/CT were analyzed. Future research should focus on evaluating the cost-effectiveness of these imaging modalities in detecting peritoneal metastases and their impact on treatment decision making.

## 5. Conclusions

Peritoneal metastases represent a common finding in women with primary or recurrent OC. Preoperative diagnostic work-up with MDCT, MRI, or FDG PET/CT is mandatory to define the extent of the disease, predict the likelihood of optimal cytoreduction, identify potentially unresectable or difficult disease locations, requiring surgical technique modifications, and select patients who may benefit from adjuvant chemotherapy.

Based on the results of this meta-analysis, FDG PET/CT and MRI had a higher diagnostic performance in the detection of PMs compared to MDCT on a per-patient analysis. No differences between the three imaging modalities were found on a per-lesion basis.

In summary, while FDG PET/CT and MRI can be considered equivalent alternatives for the detection of peritoneal metastases in ovarian cancer, the limitations of the included studies and the need for standardization should be considered. Future research addressing these limitations and exploring cost-effectiveness would contribute to the improvement in clinical practice in the management of ovarian cancer.

## Figures and Tables

**Figure 1 cancers-16-01467-f001:**
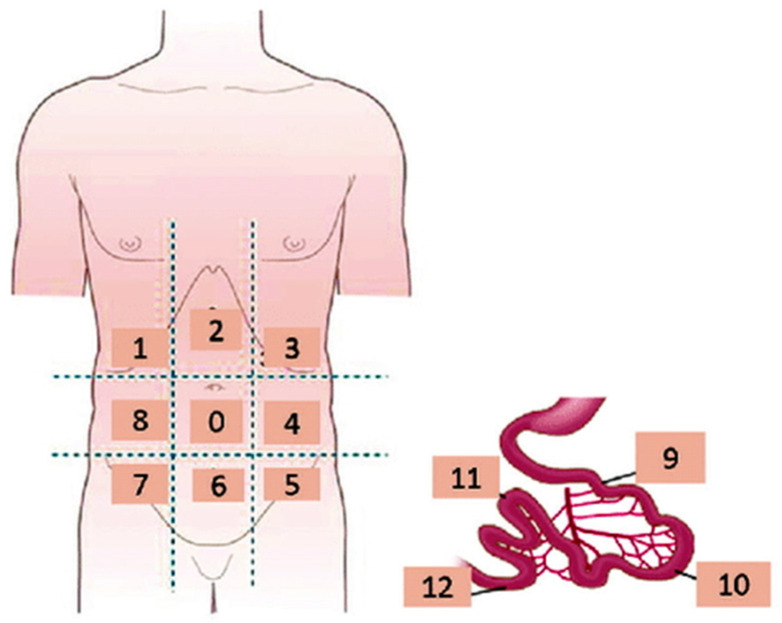
Coronal schematic drawing showing the Sugarbaker peritoneal carcinomatosis index.

**Figure 2 cancers-16-01467-f002:**
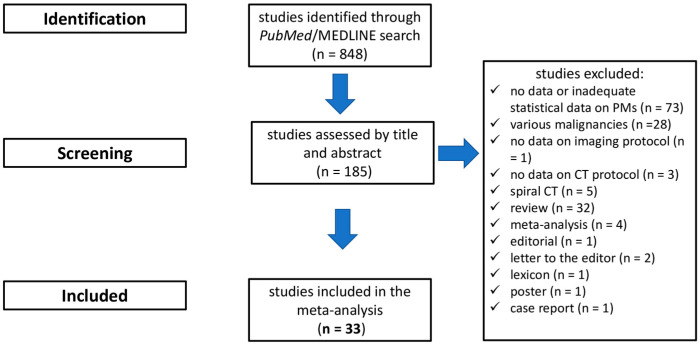
Flowchart depicting study selection.

**Figure 3 cancers-16-01467-f003:**
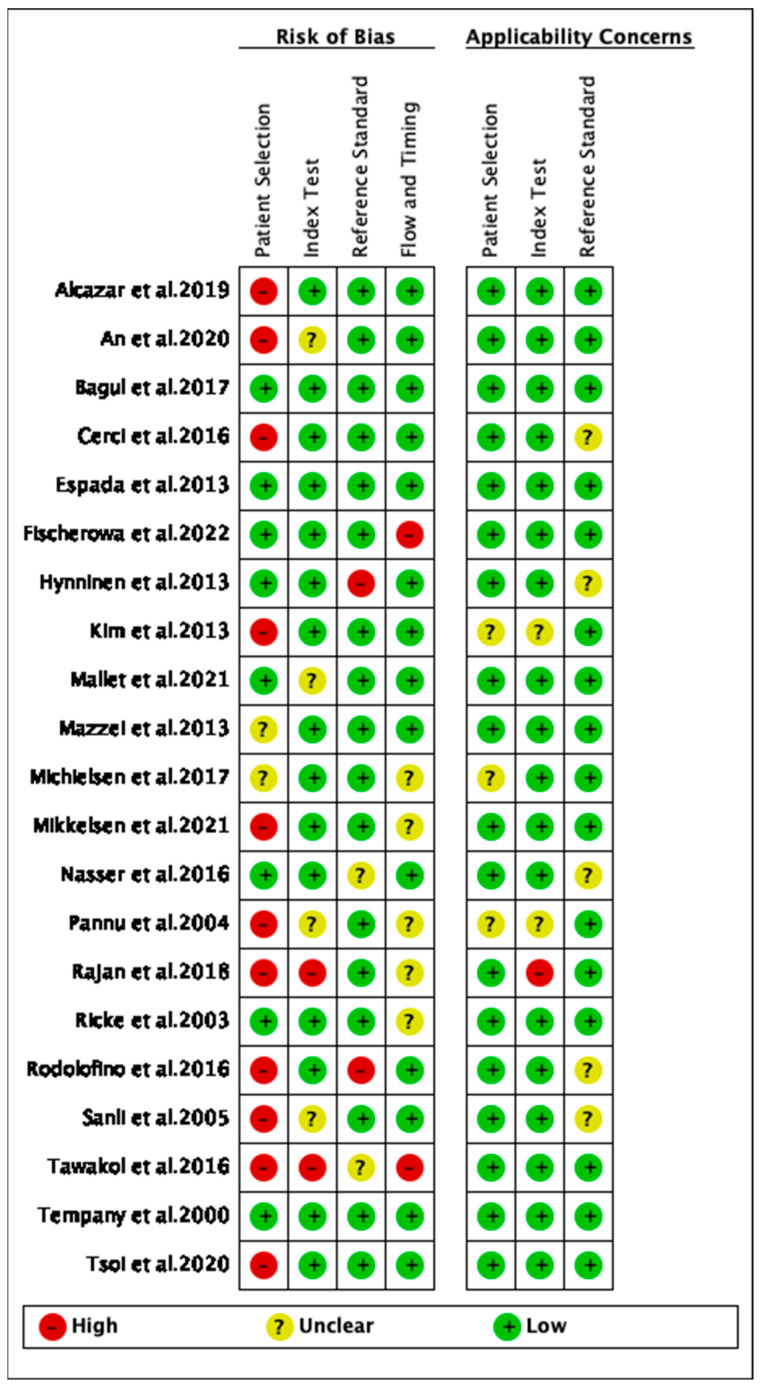
The quality assessment of the included studies [26,27,44,50,51,52,53,61,62,63,64,65,66,69,70,71,72,73,74,75,77,78].

**Figure 4 cancers-16-01467-f004:**
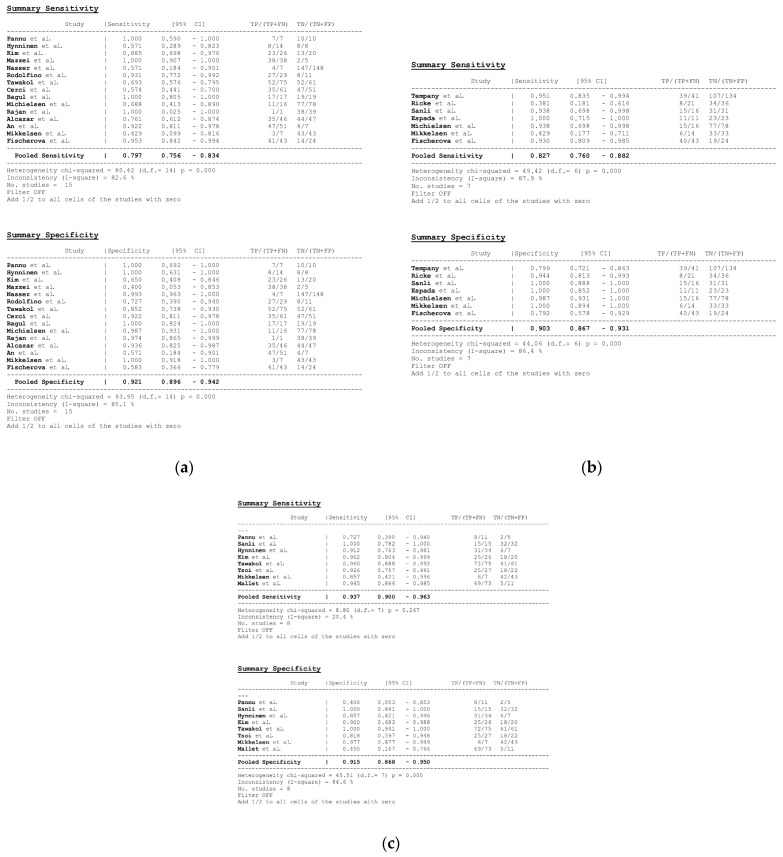
Forest plots for the pooled sensitivity and specificity calculation for (**a**) MDCT [26,27,30,52,63,64,65,69,70,71,72,73,74,75,78], (**b**) MRI [30,51,53,61,62,73,78], and (**c**) FDG PET/CT [30,44,54,61,63,64,70,77] in the detection of peritoneal metastases in ovarian cancer, on a per-patient basis (CI: confidence interval; TP: true positive; FN: false negative; TN: true negative).

**Figure 5 cancers-16-01467-f005:**
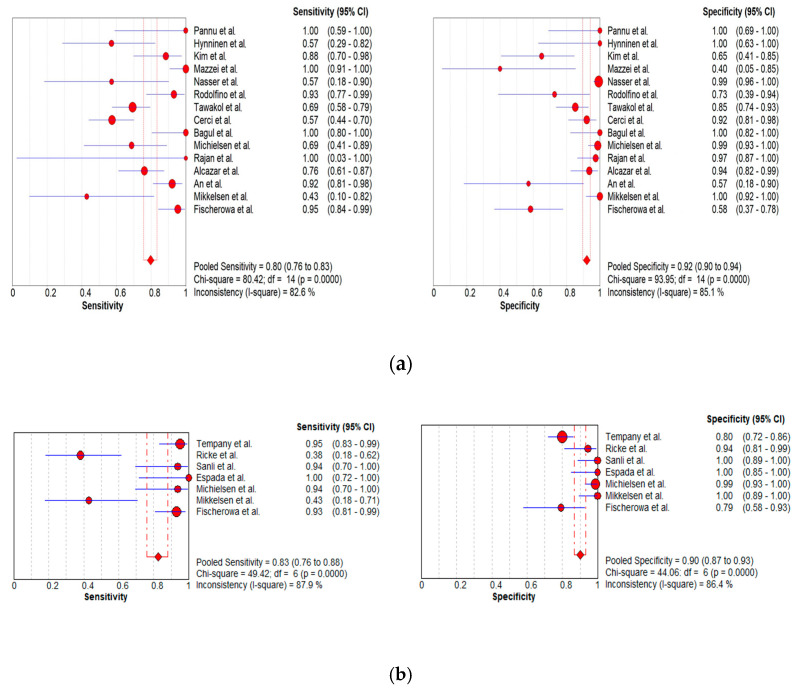
Foster plots for sensitivity and specificity for (**a**) MDCT [26,27,30,52,63,64,65,69,70,71,72,73,74,75,78], (**b**) MRI [30,44,54,61,63,64,70,77], and (**c**) FDG PET/CT [30,51,53,61,62,73,78] in the detection of peritoneal metastases in ovarian cancer, on a per-patient basis (CI: confidence interval).

**Figure 6 cancers-16-01467-f006:**
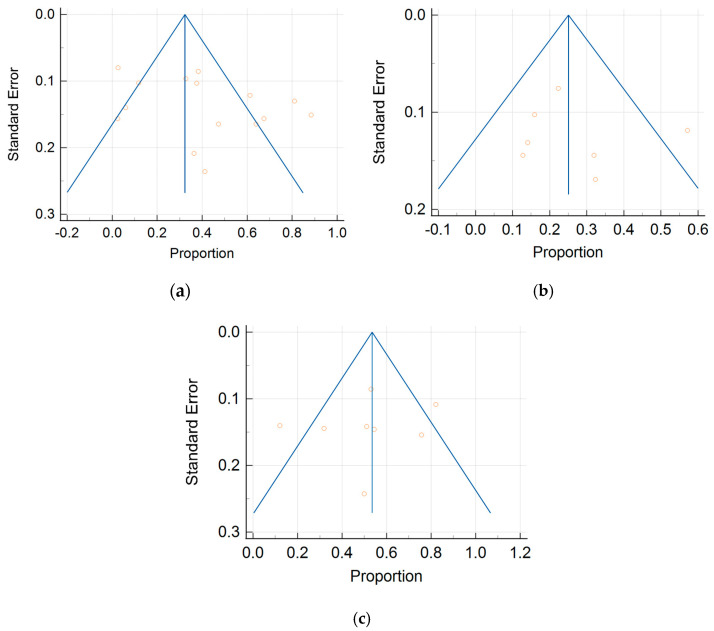
Funnel plots of the (**a**) MDCT, (**b**) MRI, and (**c**) FDG PET/CT data on a per-patient basis.

**Figure 7 cancers-16-01467-f007:**
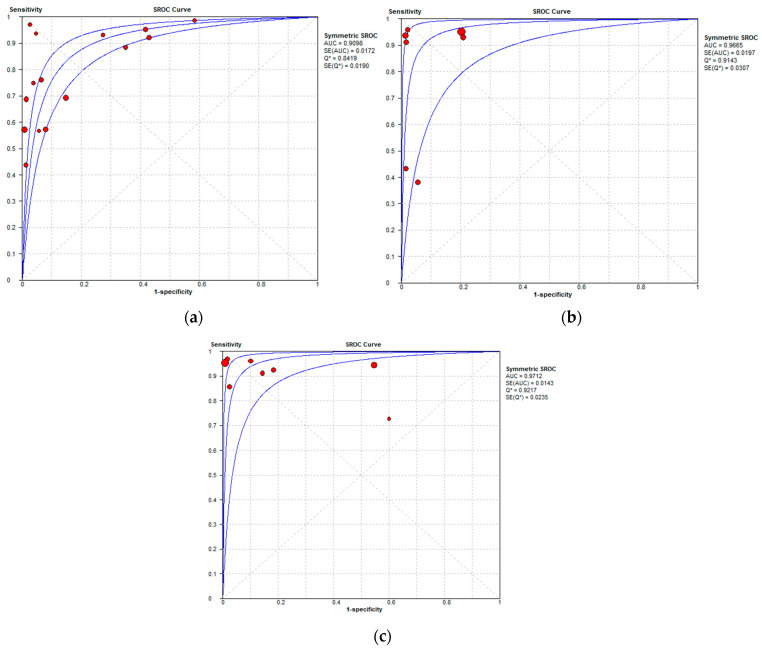
Summary receiver operating characteristic curves depicting the diagnostic performance of (**a**) MDCT, (**b**) MRI, and (**c**) FDG PET/CT in the detection of peritoneal metastases in ovarian cancer, on a per-patient basis.

**Figure 8 cancers-16-01467-f008:**
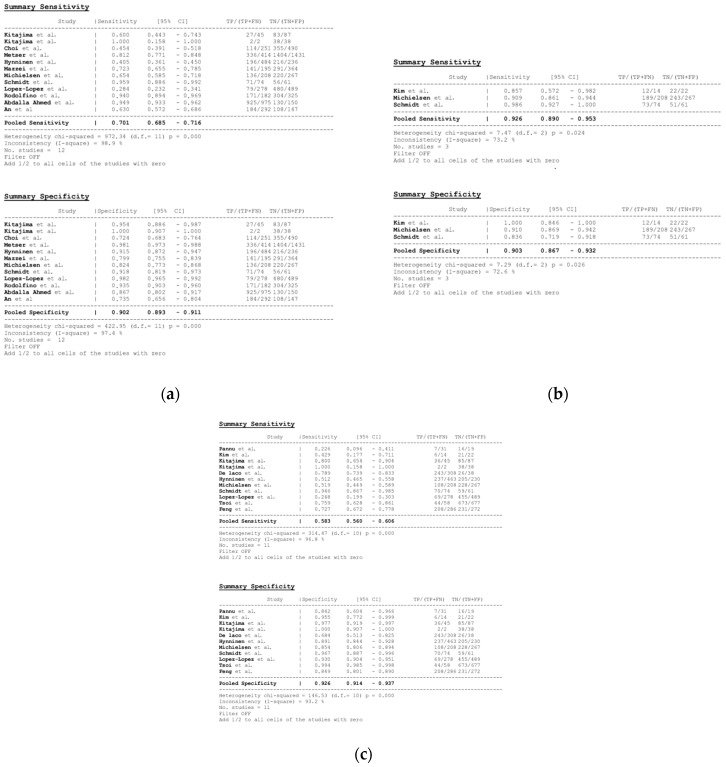
Forest plots for the pooled sensitivity and specificity calculation for (**a**) MDCT [27,56,57,58,59,63,65,66,67,68,69,76], (**b**) MRI [55,66,67], and (**c**) FDG PET/CT [42,54,55,56,57,60,63,66,67,68,77] in the detection of peritoneal metastases in ovarian cancer, on a per-region basis (CI: confidence interval; TP: true positive; FN: false negative; TN: true negative).

**Figure 9 cancers-16-01467-f009:**
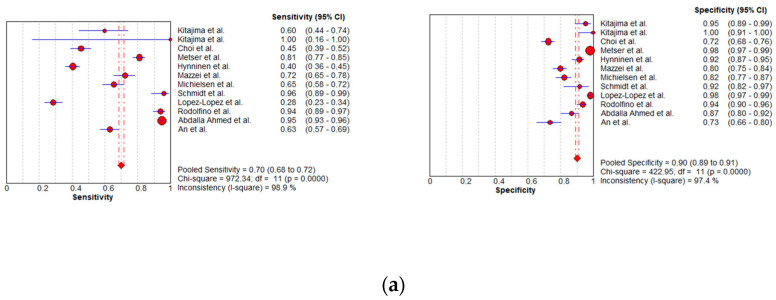
Foster plots for sensitivity and specificity for (**a**) MDCT [27,56,57,58,59,63,65,66,67,68,69,76], (**b**) MRI [55,66,67], and (**c**) FDG PET/CT [42,54,55,56,57,60,63,66,67,68,77] in the detection of peritoneal metastases in ovarian cancer, on a per-region basis (CI: confidence interval).

**Figure 10 cancers-16-01467-f010:**
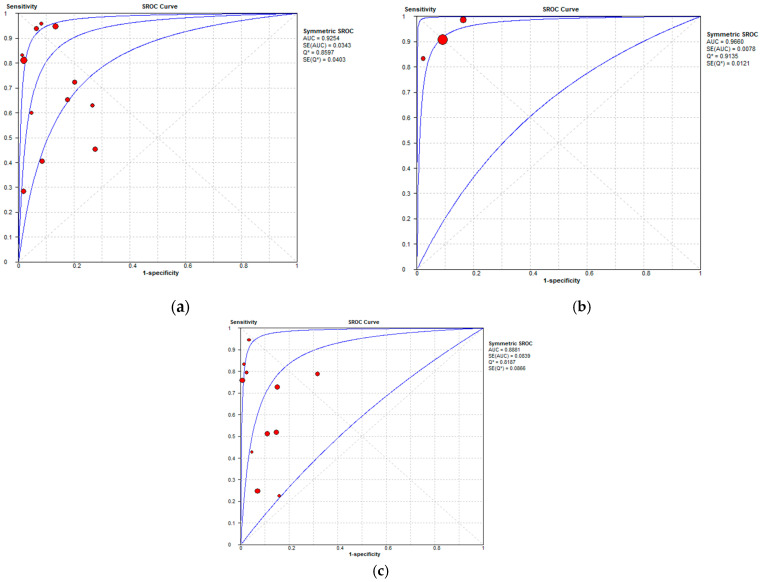
Summary receiver operating characteristic curves depicting the diagnostic performance of (**a**) MDCT, (**b**) MRI, and (**c**) FDG PET/CT in the detection of peritoneal metastases in ovarian cancer, on a per-region basis.

**Table 1 cancers-16-01467-t001:** The Sugarbaker Peritoneal Carcinomatosis Index (ARs: abdominopelvic regions; PCI: peritoneal carcinomatosis index) [23].

ARs	Sugarbaker’s PCI
AR0	midline abdominal incision, greater omentum, and transverse colon
AR1	superior surface of the right lobe of the liver, undersurface of the right hemidiaphragm, and right retrohepatic space
AR2	epigastric fat pad, left lobe of the liver, lesser omentum, and falciform ligament
AR3	undersurface of the left hemidiaphragm, spleen, pancreatic tail, and anterior and posterior surfaces of the stomach
AR4	descending colon and left paracolic gutter
AR5	pelvic side wall lateral to the sigmoid colon and sigmoid colon
AR6	female internal genitalia with ovaries, tubes and uterus, urinary bladder, cul-de-sac of Douglas, and rectosigmoid colon
AR7	right pelvic side wall and base of the cecum, including the appendix
AR8	right paracolic gutter and ascending colon
AR9–12	small bowel (AR9: upper jejunum; AR10: lower jejunum; AR11: upper ileum; and AR12: lower ileum)

**Table 2 cancers-16-01467-t002:** Characteristics of the eligible studies (OC: ovarian cancer; PMs: peritoneal metastases, ARs: abdominopelvic regions; n/a: non-applicable).

Author	Year	Type of Study	Primary Outcome	No of pts with OC	Mean Age/Age Range (Years)	FIGO Stage (No. of pts)	No. of pts with PMs	No. of ARs with PMs	Mean Size-Size Range of PMs (cm) (No. of PMs)
Tempany et al. [51]	2000	prospective	suspected advanced OC	118	57 (19–79)	III and IV (73)	70	250	<2 (8) >2 (57) n/a (5)
Pannu et al. [52]	2003	retrospective	suspected primary or recurrent OC/ peritoneal cancer	17	58.1 (41–84)	IB (1) III (13) IV (3)	13	63	n/a
Ricke et al. [53]	2003	prospective	suspected primary or recurrent OC	57	58 (35–90)	I (11) II (2) III (36) IV (4) n/a (4)	n/a	204	n/a
Pannu et al. [54]	2004	retrospective	suspected recurrent OC	16	50.8 (17–77)	n/a	11	31	<1 (23) >1 (8)
Kim et al. [55]	2007	retrospective	suspected recurrent OC	36	51.3 (25–75)	I (2) II (5) III (27) IV (2)	n/a	14	2.2 (0.4–3.5)
Kitajima et al. [56]	2008	retrospective	suspected recurrent OC	132	56 (34–79)	Ι (20) II (10) III (81) IV (21)	n/a	45	n/a
Kitajima et al. [57]	2008	retrospective	primary OC	40	55.4 (38–77)	Ι (18) II (7) III (14) IV (1)	n/a	46	0.2–2.3
Choi et al. [58]	2011	prospective	primary OC	57	53.1 (30–72)	I (6) II (5) III (38) IV (8)	50	251	<1 >1
Metser et al. [59]	2011	retrospective	primary OC	76	58.2 (24–87)	I (11) II (3) III (55) IV (7)	n/a	414	<1 (142) ≥1 (272)
De laco et al. [60]	2011	retrospective	suspected OC	40	65 ± 7.9 (46–78)	III (22) IV (18)	40	308	≤0.5 (135) 0.5–5 (38) >5 (135)
Sanli et al. [61]	2012	retrospective	suspected recurrent OC	47	57.5 ± 8.4 (38–78)	n/a	n/a	n/a	<0.5 0.5–1 1–2 2–3 >3
Espada et al. [62]	2013	prospective	suspected advanced OC	34	53.08 ± 11.9	III (28) IV (6)	n/a	n/a	n/a
Hynninen et al. [63]	2013	prospective	suspected advanced ovarian/ fallopian/ peritoneal cancer	41	65 (45–79)	I (2) II (2) III (21) IV (16)	41	246	n/a
Kim et al. [64]	2013	retrospective	suspected primary or recurrent OC	46	54 (29–80)	I (12) II (4) III (28) IV (2)	26	n/a	n/a
Mazzei et al. [65]	2013	retrospective	advanced primary or recurrent OC	43	58.5 (30–72)	III (42) IV (1)	43	195	<0.5 0.5–5 >5
Michielsen et al. [66]	2014	prospective	suspected OC	32	61.9 (20–83)	n/a	32	208	<1 (75) >1 (60) confluent disease (73)
Schmidt et al. [67]	2015	prospective	suspected OC	15	65 (31–89)	III (4) IV (6)	10	74	≤0.5 (13) 0.5–5 (40) >5 cm (21)
Lopez-Lopez et al. [68]	2016	retrospective	suspected primary or recurrent OC	59	54 (27–78)	I (3) II (44) III (12)	55	278	<0.5 (110) ≥0.5–5 (53) >5 cm or confluent (115)
Nasser et al. [26]	2016	retrospective	suspected primary or recurrent OC	155	62.5 (31–85)	I (4) II (3) III (106) IV (42)	n/a	n/a	n/a
Rodolfino et al. [69]	2016	retrospective	suspected recurrent OC	40	48.5 (32–73)	III (33) IV (7)	29	182	<0.5 (38) ≥0.5–5 (81) >5 cm or confluent (63)
Tawakol et al. [70]	2016	prospective	suspected recurrent OC	111	54 (13–76)	n/a	n/a	75	n/a
Cerci et al. [71]	2016	retrospective	primary OC	114	59 (28–91)	I (21) II (4) III (47) IV (39)	n/a	n/a	n/a
Bagul et al. [72]	2017	prospective	suspected advanced ovarian/ fallopian tube/primary peritoneal cancer	36	51 (39–74)	IIIc	n/a	n/a	n/a
Michielsen et al. [73]	2017	prospective	suspected OC	94	61 (14–88)	I (19) II (2) III (38) IV (35)	n/a	n/a	n/a
Rajan et al. [74]	2018	prospective	advanced OC	40	59.5 (43–87)	IIIc IV	40	115	<0.5 0.5–5 >5
Alcazar et al. [75]	2019	retrospective	suspected OC	93	57.6 ± 11.4 (18–84)	I (26) II (11) IIIA (1) IIIB (6) IIIC (40) IVA (6) IVB (3)	n/a	n/a	n/a
Abdalla Ahmed et al. [76]	2019	prospective	primary OC	85	55 (27–82)	II (5) III (80)	n/a	930	8.4 (1–13) <0.5 (280) 0.5–5 (605) >5 (45)
Tsoi et al. [77]	2020	retrospective	primary or recurrent ovarian/peritoneal cancer	49	49 ± 15	I (15) II (12) III (18) IV (1) n/a (3)	27	58	<1 (9) ≥1 (44)
An et al. [27]	2020	retrospective	recurrent advanced OC	58	57 (23–84)	III (31) IV (27)	n/a	315	3.7 (1–15)
Mikkelsen et al. [30]	2021	prospective	advanced OC	50	65 (32–78)	III (32) IV (18)	n/a	n/a	n/a
Feng et al. [42]	2021	prospective	advanced OC	43	57 (38–76)	III (32) IV (11)	n/a	286	n/a
Mallet et al. [44]	2021	retrospective	advanced OC	84	65 (44–89)	III (28) IV (56)	n/a	n/a	<0.5 0.5–5 >5
Fischerova et al. [78]	2022	prospective	suspected primary advanced ovarian/tubal/peritoneal cancer	67	61.4 ± 10.5	I (14) II (2) III (44) IV (7)	n/a	n/a	n/a

**Table 3 cancers-16-01467-t003:** Location of PMs (ARs: abdominopelvic regions; n/a: non-applicable).

Study	Location of PMs (*n* = Number of Patients or ARs with PMs)
Tempany et al. [51] (*n* = number of ARs)	anterior part of the abdomen (37) RT, LT paracolic gutters AR4,8 (35) RT, LT subdiaphragmatic spaces AR1,3 (45) mesentery (small bowel/transverse/sigmoid colon) (38) hepatic surface AR1,2 (25) omentum (gastrocolic and infracolic) AR0 (70)
Pannu et al. [52] (*n* = number of patients)	diaphragm AR1,3 (11) liver AR1,2 (6) splenic surface AR3 (2) porta hepatis/gallbladder fossa AR1 (4) stomach AR3 (2) lesser sac AR2 (3) mesenteric root (3) infracolic omentum AR0 (7) paracolic gutters AR4,8 (8) bowel (5) pelvis AR5–7 (12)
Ricke et al. [53] (*n* = number of ARs)	pouch of Douglas AR6 (18) cervix/vaginal stump AR6 (9) uterus AR6 (11) bladder/ureter AR6 (10) pelvic wall AR5,7 (23) abdominal wall (18) small bowel/mesentery (22) large bowel (39) greater omentum AR0 (21) lesser sac AR2 (7) stomach AR3 (6) diaphragm AR1,3 (15) liver capsule AR1,2 (5)
Pannu et al. [54] (*n* = number of ARs)	pelvis AR5–7 (17) bowel/omentum (7) LT upper quadrant AR3 (2) paracolic gutters AR4,8 (4) RT upper quadrant AR1 (1)
Kim et al. [55] (*n* = number of ARs)	cul de sac AR6 (4) paracolic gutter AR4,8 (3) subphrenic/perihepatic/perisplenic (6) bowel (1)
Kitajima et al. [56] (n/a)	cul de sac AR6 paracolic gutter AR4,8 mesentery serosa of large and small bowel anterior part of the abdomen hepatic surface AR1,2 splenic hilum AR3 diaphragm AR1,3
Kitajima et al. [57] (*n* = number of ARs)	cul de sac AR6 (8) urinary bladder AR6 (2) rectosigmoid colon AR6 (4) peritoneum of anterior abdomen (6) paracolic gutter AR4,8 (3) diaphragm AR1,3 (1) omentum AR0 (9) mesentery (7) serous membrane of large and small bowel (4) liver surface AR1,2 (2)
Choi et al. [58] (*n* = number of ARs)	RT subdiaphragmatic area AR1 (35) LT subdiaphragmatic area AR3 (34) porta hepatis AR1 (10) lesser sac AR2 (18) small bowel mesentery (14) splenic hilar area AR3 (38) omentum AR0 (20) RT paracolic gutter AR8 (22) LT paracolic gutter AR4 (20) RT pelvic cavity AR7 (4) LT pelvic cavity AR5 (8) sigmoid mesentery (12) bladder dome area AR6 (16)
Metser et al. [59] (n/a)	RT diaphragm AR1 liver capsule AR1,2 liver parenchymal invasion gallbladder fossa AR1 RT paracolic gutter AR8 LT diaphragm AR3 omentum AR0 LT paracolic gutter AR4 bladder peritoneum AR6 porta hepatis AR1 root of small bowel mesentery mesentery ascending colon, serosa AR8 cecum, serosa AR7 appendix AR7 stomach, serosa AR3 small bowel, serosa transverse colon, serosa AR0 descending colon, serosa AR4 spleen, capsule AR3 spleen, hilum AR3 spleen, parenchymal invasion rectosigmoid mesentery RT pelvic sidewall AR7 LT pelvic sidewall AR5 cul-de-sac, posterior AR6 rectosigmoid, serosa AR6 rectosigmoid, invasion AR6
De laco et al. [60] (*n* = number of ARs)	central AR0 (37) RT upper AR1 (34) epigastrium AR2 (26) LT upper AR3 (28) LT flank AR4 (35) LT lower AR5 (36) pelvis AR6 (40) RT lower AR7 (38) RT flank AR8 (34)
Espada et al. [62] (n/a)	small and/or large bowel mesentery (8) hepatic parenchyma, hepatic hilum or surface implants > 2 cm AR1,2 (10) omental extension: spleen parenchyma, splenic hilum, stomach, lesser sac AR2,3 (11) diaphragm AR1,3 (5) peritoneal
Hynninen et al. [63] (*n* = number of patients)	diaphragm AR1,3 (34) omentum AR0 (34) small bowel mesentery (25) large bowel mesentery (30) small bowel serosae AR9–12 (14) large bowel serosae (64) RT ‘high risk upper abdomen’: dorsal subdiaphragmatic peritoneum, dorsal liver surface AR1 (31) LT ‘high risk upper abdomen’: ventricle, bursa omentalis, spleen, tail of pancreas AR3 (14)
Mazzei et al. [65] (n/a)	central AR0 RT upper AR1 epigastrium AR2 LT upper AR3 LT flank AR4 LT left AR5 pelvis AR6 RT lower AR7 RT flank AR8 upper jejunum AR9 lower jejunum AR10 upper ileum AR11 lower ileum AR12
Michielsen et al. [66] (*n* = number of ARs)	bladder peritoneal surface AR6 (17) Douglas pouch AR6 (19) RT peritoneal pelvic surface AR7 (20) RT lateroconal area AR8 (15) subhepatic space/Morrison’s pouch AR1 (10) RT diaphragm AR1 (12) hepatic surface AR1,2 (4) LT diaphragm AR3 (9) splenic surface AR3 (1) LT lateroconal area AR4 (16) LT peritoneal pelvic surface AR5 (21) omentum AR0 (23) small bowel serosa AR9–12 (6) small bowel mesentery (12) colonic serosa (11) colonic mesentery (12)
Schmidt et al. [67] (n/a)	central AR0 RT upper AR1 epigastrium AR2 LT upper AR3 LT flank AR4 LT lower AR5 pelvis AR6 RT lower AR7 RT flank AR8
Lopez-Lopez et al. [68] (n/a)	upper region middle region lower region small intestine
Nasser et al. [26] (*n* = number of patients)	diaphragmatic involvement AR1,3 (55) splenic involvement AR3 (19) large bowel involvement (37) small bowel involvement (15) rectal involvement AR6 (38) porta hepatis involvement AR1 (6) mesenteric involvement (35)
Rodolfino et al. [69] (n/a)	central AR0 RT upper AR1 epigastrium AR2 LT upper AR3 LT flank AR4 LT lower AR5 pelvis AR6 RT lower AR7 RT flank AR8 upper jejunum AR9 lower jejunum AR10 upper ileum AR11 lower ileum AR12
Cerci et al. [71] (*n* = number of patients)	peritoneal carcinomatosis (61) omentum AR0 (53) ascites (61) perivesical-perirectal fat AR6 (54) diaphragm AR1,3 (20) liver AR1,2 (30) bladder AR6 (22) small and large bowel (47) mesentery (49)
Bagul et al. [72] (*n* = number of patients)	diffuse peritoneal thickening (17) RT subdiaphragm AR1 (35) LT subdiaphragm AR3 (27) porta hepatis AR1 (24) liver AR1,2 (25) spleen AR3 (11) lesser sac AR2 (15) omentum AR0 (35) omental cake extension (to splenic hilum, stomach, colon, or lesser sac) (24) RT paracolic region AR8 (33) LT paracolic region AR4 (27) small bowel serosa (20) large bowel serosa (29) small bowel mesentery (21) large bowel mesentery (32) uterus and ovary AR6 (34) pelvic peritoneum AR5–7 (34) urinary bladder peritoneum AR6 (32) parietal peritoneum (19)
Michielsen et al. [73] (*n* = number of patients)	duodenum, stomach, celiac trunk carcinomatosis AR2 (16) diffuse serosal carcinomatosis (34) superior mesenteric artery, mesenteric root (8)
Rajan et al. [74] (*n* = number of patients)	central AR0 (24) RT upper AR1 (4) epigastrium AR2 (6) LT upper AR3 (1) LT flank AR4 (7) LT lower AR5 (12) pelvis AR6 (36) RT lower AR7 (5) RT flank AR8 (8) upper jejunum AR9 (3) lower jejunum AR10 (2) upper ileum AR11 (2) lower ileum AR12 (5)
Alcazar et al. [75] (*n* = number of patients)	rectosigmoid AR6 (27) pelvic peritoneum AR5–7 (59) major omentum AR0 (46) upper abdominal peritoneum (43) small bowel (12) mesentery (4) mesogastrium AR2 (12) hepatic hilum AR1 (10) spleen AR3 (5)
Abdalla Ahmed et al. [76] (n/a)	central AR0 RT upper AR1 LT upper AR3 LT flank AR4 LT lower AR5 pelvis AR6 Douglas pouch, rectosigmoid colon AR6 RT lower AR7 RT flank AR8 upper jejunum AR9 lower jejunum AR10 upper ileum AR11 lower ileum AR12
Tsoi et al. [77] (*n* = number of patients)	RT subphrenic space AR1 (3) RT subhepatic space AR1 (2) gastric serosa AR2 (1) lesser sac AR2 (0) LT subphrenic space AR3 (1) LT perihepatic space AR2 (0) RT paracolic gutter AR8 (2) LT paracolic gutter AR4 (2) pouch of Douglas AR6 (6) bladder flap AR6 (6) mesentery (5) omentum AR0 (6) large bowel serosa (9) small bowel serosa (1) pelvis AR5–7 (14)
An et al. [27] (*n* = number of ARs)	subdiaphragmatic space AR1,3 (24) perihepatic space/Morrison pouch AR1 (24) porta hepatis AR1 (2) upper abdominal peritoneum/stomach serosa, lesser sac AR2 (33) splenic hilum AR3 (2) paracolic gutters AR4,8 (27) bowel serosa (45) bowel mesentery (35) omentum AR0 (48) pelvic peritoneum AR5–7 (75)
Mikkelsen et al. [30] (*n* = number of patients)	liver/duodenum/pancreas/gastric ventricle (7) porta hepatis/hepatoduodenal ligament AR1 (17) celiac trunk/superior mesenteric artery/bowel mesentery root (47)
Feng et al. [42] (*n* = number of ARs)	central AR0 (29) RT upper AR1 (31) epigastrium AR2 (11) LT upper AR3 (16) LT flank AR4 (19) LT lower AR5 (32) pelvis AR6 (43) RT lower AR7 (29) RT flank AR8 (21) upper jejunum AR9 (10) lower jejunum AR10 (10) upper ileum AR11 (18) lower ileum AR12 (17)
Mallet et al. [44] (*n* = number of patients)	central AR0 (73) RT upper AR1 (67) epigastrium AR2 (58) LT upper AR3 (56) LT flank AR4 (58) LT lower AR5 (77) pelvis AR6 (78) RT lower AR7 (71) RT flank AR8 (62) upper jejunum AR9 (19) lower jejunum AR10 (22) upper ileum AR11 (31) lower ileum AR12 (35)
Fischerova et al. [78] (n/a)	pelvic involvement: anterior and posterior compartment AR5–7 rectosigmoid AR6 upper abdominal involvement: LT diaphragm, spleen, RT diaphragm, liver, and lesser omentum greater omentum: supracolic and infracolic omentum AR0 colon infiltration by omentum RT and LT paracolic gutter AR4,8 anterior abdominal wall bowel serosal and mesenterial peritoneal involvement: small and large bowel serosa and small and large bowel mesentery

**Table 4 cancers-16-01467-t004:** Characteristics of imaging modalities (MDCT: multidetector CT; DWI: diffusion-weighted imaging; 18F FDG-PET/CT: fluorine-18-fluorodeoxyglucose positron emission tomography/CT, PDS: primary debulking surgery; SLL: second-look laparotomy; IDS: interval debulking surgery; EL: exploratory laparotomy; CECT: contrast-enhanced CT, WB-MRI: whole-body MRI; n/a: non-applicable).

Study	Standard of Reference	MDCT (*n* = 23)	MRI (*n* = 4)	DWI (*n* = 6)	18F FDG-PET/CT (*n* = 16)	Mean Time Interval between Imaging Modalities/Range (Days)	Mean Time Interval between Imaging and Surgery/Range (Days)
Tempany et al. [51]	surgical (PDS) and histopathologic findings		YES			-	28
Pannu et al. [52]	surgical findings (PDS or SLL)	YES				-	16 (2–108)
Ricke et al. [53]	surgical (laparotomy) and histopathologic findings		YES			-	56
Pannu et al. [54]	surgical (laparotomy) and histopathologic findings				YES	-	31.7 (6–110)
Kim et al. [55]	surgical and/or histopathologic findings (SLL or biopsy), radiological and clinical follow-up		YES		YES	10 (1–20)	18 (2–35)
Kitajima et al. [56]	surgical and/or histopathologic findings (SLL or biopsy), radiological and clinical follow-up of at least 6 months	YES			YES (CECT)	concurrent	n/a
Kitajima et al. [57]	surgical (PDS) and histopathologic findings	YES			YES (CECT)	concurrent	14
Choi et al. [58]	surgical findings (PDS)	YES				-	17.6 (2–44)
Metser et al. [59]	surgical (PDS or IDS) and histopathologic findings, follow-up (mean time: 19 months)	YES				-	24 (1–67)
De laco et al. [60]	surgical (laparoscopy) and histopathologic findings				YES	-	n/a
Sanli et al. [61]	surgical and histopathologic findings (surgical exploration or biopsy), clinical follow-up of at least 6 months		YES		YES	≤30	n/a
Espada et al. [62]	surgical (EL) and histopathologic findings			YES		-	15
Hynninen et al. [63]	surgical (PDS, laparotomy or laparoscopy + IDS) and histopathologic findings	YES			YES (CECT)	concurrent	14
Kim et al. [64]	surgical (PDS or IDS) and histopathologic findings	YES			YES	17 (1–60)	PET/CT: 23 (1–54) MDCT: 26 (4–61)
Mazzei et al. [65]	surgical (PDS) and histopathologic findings	YES				-	45
Michielsen et al. [66]	surgical (PDS or IDS) and histopathologic findings, imaging follow-up	YES		YES (WB-MRI)	YES (CECT)	n/a	n/a
Schmidt et al. [67]	surgical and histopathologic findings	YES		YES	YES	1 ± 4 (0–14)	8.1 ± 2.4 (1–29)
Lopez-Lopez et al. [68]	surgical findings	YES			YES	n/a	<42
Nasser et al. [26]	surgical (debulking surgery) and histopathologic findings	YES				-	n/a
Rodolfino et al. [69]	imaging follow-up for a minimum of 12 months	YES				-	n/a
Tawakol et al. [70]	surgical and histopathologic findings (surgical exploration, biopsy), imaging and clinical follow-up for at least 6 months	YES			YES (CECT)	concurrent	n/a
Cerci et al. [71]	surgical and histopathologic findings	YES				-	28
Bagul et al. [72]	surgical (PDS) and histopathologic findings	YES				-	14
Michielsen et al. [73]	surgical (PDS or IDS) and histopathologic findings, imaging follow-up	YES		YES (WB-MRI)		n/a	n/a
Rajan et al. [74]	surgical (PDS or IDS) and histopathologic findings	YES				-	n/a
Alcazar et al. [75]	surgical and histopathologic findings (surgical exploration, biopsy)	YES				-	15
Abdalla Ahmed et al. [76]	surgical (laparoscopy and laparotomy, PDS) and histopathologic findings	YES				-	10 (12 ± 5)
Tsoi et al. [77]	surgical (debulking surgery) and histopathologic findings				YES (CECT)	-	19 ± 16
An et al. [27]	surgical (IDS) and histopathologic findings or imaging follow-up in 6–12 months	YES				-	13 (2–43)
Mikkelsen et al. [30]	surgical (PDS) and histopathologic findings	YES (PET/CT)		YES	YES (CECT)	n/a	DWI: 15 (6–28) PET/CT: 14 (1–27)
Feng et al. [42]	surgical (PDS) and histopathologic findings				YES	-	14
Mallet et al. [44]	surgical (laparoscopy) and histopathologic findings				YES	-	28
Fischerova et al. [78]	surgical (laparoscopy or laparotomy, PDS) and histopathologic findings	YES		YES (WB-MRI)		few	28

**Table 5 cancers-16-01467-t005:** Description of MDCT features (n/a: non-applicable; cm: contrast medium; mgI/mL: iodine content; kV: kilovolt; MIP: maximum-intensity projection; CECT: contrast-enhanced CT).

Summary of MDCT Features									
Study	Number of Rows	Type of Intravenous cm (mgI/mL)	Amount of cm	Type of Luminal cm	Phases	Slice Thickness (mm)	Slice Reconstruction (mm)	kV	MPRs
Pannu et al. [52]	4	non-ionic	120 mL	750–1000 mL H_2_O	arterial, portal	3	2	n/a	coronal, sagittal, oblique
Kitajima et al. [PET/CECT] [56]	16	Iomeprole 300	2 mL/kg (150 mL max)	No	portal	2	n/a	140	coronal, sagittal
Kitajima et al. PET/CECT] [57]	16	Iomeprole 300	2 mL/kg (150 mL max)	No	portal	2	n/a	140	coronal, sagittal
Choi et al. [58]	4	Ultravist 300	140 mL	n/a	portal	3.2	3	n/a	n/a
Metser et al. [59]	64	Omnipaque 300	2 mL/kg (180 mL max)	n/a	portal	5	2	120	coronal
Hynninen et al. [PET/CECT] [63]	64	Yes, n/a	n/a	n/a	n/a	n/a	n/a	120	n/a
Kim et al. [64]	16 or 64	Yes, n/a	130 mL	450 mL n/a	n/a	n/a	3	120	n/a
Mazzei et al. [65]	4 or 16 or 64	Iopamiro 370	2 mL/kg	H_2_O + Macrogol (7 patients)	late arterial, portal	3.75 (4-row) 3.75/2.5 (16-row) 3.75/1.25/2.5 (64-row)	1.5 (4-row) 0.8 (16-row) 0.8 (64-row)	120–140	coronal, sagittal, oblique
Michielsen et al. [66]	16 or 64	Visipaque 320	120 mL	30 mL Telebrix + 900 mL H_2_O	portal	5	n/a	120	n/a
Schmidt et al. [67]	64	Iohexol 300	body weight + 30 mL	1 L H_2_O (rectal enema)	portal	2	2	120	Yes, n/a
Lopez-Lopez et al. [68]	n/a	Yes, n/a	130 mL	450 mL n/a	portal	n/a	3	120	n/a
Nasser et al. [26]	n/a	n/a	n/a	n/a	n/a	n/a	n/a	n/a	n/a
Rodolfino et al. [69]	n/a	Iopromide 370	2 mL/kg	Gastrografin 15 mL + 300 mL H_2_O	portal, delayed	1	n/a	n/a	Yes, n/a
Tawakol et al. [PET/CECT] [70]	64	non-ionic	1–2 mL/kg (150 mL max)	400–600 mL diluted mannitol	n/a	1.5	n/a	120	axial, coronal, sagittal, MIP
Cerci et al. [71]	n/a	Yes, n/a	n/a	Yes, n/a	n/a	n/a	n/a	n/a	n/a
Bagul et al. [72]	64	non-ionic	80 mL	Gastrografin 2%, 40 mL + 2 L H_2_O	arterial	3–5	n/a	n/a	n/a
Michielsen et al. [73]	16 or 64	Yes, n/a	n/a	Yes, n/a	portal	n/a	3–5	n/a	transverse, coronal
Rajan et al. [74]	16	non-ionic	50 mL	1000 mL diluted contrast 2%	arterial, portal	5	2–3	n/a	n/a
Alcazar et al. [75]	64	n/a	n/a	n/a	n/a	n/a	n/a	n/a	n/a
Abdalla Ahmed et al. [76]	16 or 64	Ultravist 300	140 mL	500–750 mL n/a	arterial, portal	1.25	0.8	120	n/a
An et al. [27]	64	n/a	1.5 mL/kg	No	portal	2.5	2.5	120	coronal, sagittal
Mikkelsen et al. [30]	64	Iomeron	0.8 mL/kg	diluted Omnipaque	n/a	n/a	2.5 mm	n/a	n/a
Fischerova et al. [78]	n/a	non-ionic	n/a	1 L H_2_O or diluted iodine contrast	portal	n/a	n/a	n/a	coronal, sagittal, axial

**Table 6 cancers-16-01467-t006:** Description of MRI features (T: Tesla; cm: contrast medium; WB-DWI: whole-body diffusion-weighted imaging; im: intramuscularly; iv: intravenously; n/a: non-applicable; DCE: dynamic contrast-enhanced).

Summary of MRI Features									
Study	Magnetic Field Strength (T)	Type of Coil	Type of Intravenous cm (mg/mL)	Amount of cm	Phases	Type of Luminal Contrast	Bowel Preparation	Section Thickness (mm)	*b*-Value (s/mm^2^)
Tempany et al. [51]	1.5	multicoil array or body	Gadolinium	n/a	n/a	n/a	1 mg GlucaGen (im)	8–10	No
Ricke et al. [53]	1.5	body	Magnevist	0.2 mL/kg	n/a	n/a	2 × 20 mg Buscopan (iv)	8	No
Kim et al. [55]	1.5	phased array or body	Magnevist	0.1 mmol/kg	n/a	No	20 mg Buscopan (im)	5–8	No
Sanli et al. [61]	1.5	phased array	n/a	n/a	arterial, venous, delayed	n/a	n/a	4–8	No
Espada et al. [62]	3	phased array	Gadolinium	n/a	n/a	n/a	n/a	5	600
Michielsen et al. [66]	3	phased array	Gadolinium-DOTA	15 mL	portal	1 L pineapple juice	20 mg Buscopan (iv)	1.5–6	WB-DWI 0, 1000
Schmidt et al. [67]	3	phased array + spine clusters	Gadolinium-DOTA	0.2 mmol/kg	portal	1 L H_2_O rectal enema	20 mg Buscopan/1 mg GlucaGen (iv)	3–6	0, 300, 600
Michielsen et al. [73]	3	phased array	Gadolinium	n/a	n/a	1 L pineapple juice	20 mg Buscopan (iv)	2.5–6	WB-DWI 0, 1000
Mikkelsen et al. [30]	1.5	multi-channel	No	n/a	n/a	No	1 mg glucagon im	5–8	0, 1000
Fischerova et al. [78]	3	phased array	Gadolinium	n/a	n/a	1 L pineapple juice or H_2_O	Buscopan (iv)	5	WB-DWI 50, 1000

**Table 7 cancers-16-01467-t007:** Description of FDG PET/CT features (n/a: non-applicable; cm; contrast medium; mgI/mL: iodine content; kV: kilovolt).

Summary of FDG PET/CT Features									
PET					CT				
Study	System (Covered Area)	Tracer Amount	Scanning Time (min)	Scanning Time (min) per Bed Position	Number of Rows	Slice Thickness (mm)	Type of Intravenous cm (mgI/mL)	Type of Luminal cm	kV
Pannu et al. [54]	caudal to cranial direction	0.22 mCi/kg	n/a	5	4	n/a	No	Readi-cat 1.3%	140
Kim et al. [55]	head-pelvic floor	260–485 MBq	n/a	5	8	5	No	No	140
Kitajima et al. [56]	ear-mid thigh	4 MBq/kg	18–21	3	16	2	Iomeprole 300, 2 mL/kg (150 mL max)	No	140
Kitajima et al. [57]	ear-mid thigh	4 MBq/kg	18–21	3	16	2	Iomeprole 300, 2 mL/kg (150 mL max)	No	140
De laco et al. [60]	n/a	5.3 MBq/kg	n/a	4	n/a	5	No	n/a	120
Sanli et al. [61]	skull-upper thigh	370–550 MBq	18–24	3	n/a	n/a	No	Yes, n/a	140
Hynninen et al. [63]	skull-mid thigh	4 MBq/kg	n/a	n/a	64	n/a	Yes, n/a	n/a	120
Kim et al. [64]	skull to upper thigh	350 MBq	n/a	3	16	3.75	No	No	120
Michielsen et al. [66]	whole-body	303 MBq (220–388)	n/a	n/a	spiral	5	Yes, n/a	Yes, n/a	120
Schmidt et al. [67]	skull base-mid thigh	5.5 MBq/kg	n/a	n/a	16	5	No	n/a	140
Lopez-Lopez et al. [68]	skull base-upper thigh	370 MBq	n/a	3	16	5	No	No	120
Tawakol et al. [70]	skull base-mid thigh	3.7–5.2 MBq/kg	18	2	64	5	non-ionic, 1–2 mL/kg, 150 mL max	400–600 mL mannitol	120
Tsoi et al. [77]	skull base-proximal thigh	298 + 53 MBq	15	2.5	64	2.5	±iodinated	No	120
Mikkelsen et al. [30]	n/a	4 MBq/kg	n/a	n/a	64	2.5	Iomeron 0.8 mL/kg	dilute Omnipaque	n/a
Feng et al. [42]	inguinal region-head	7.4 MBq/kg	n/a	2–3	16	n/a	n/a	n/a	120
Mallet et al. [44]	head to midthighs	2–4 MBq/kg	n/a	n/a	n/a	n/a	n/a	n/a	n/a

**Table 8 cancers-16-01467-t008:** Diagnostic accuracy of MDCT in different abdominopelvic regions on a per-patient basis (AR: abdominopelvic region; CI: confidence interval; AUC: area under the curve).

ARs	Pooled Sensitivity (%CI)	Pooled Specificity (%CI)	AUC
AR0	80.1 (74.5–84.9)	89 (83.1–93.3)	0.91
AR1	62.5 (54.1–70.4)	97.1 (94.8–98.6)	0.86
AR2	53.1 (34.7–70.9)	92.8 (86.8–96.7)	0.72
AR3	61.8 (50.9–71.9)	97.9 (95.8–99.2)	0.93
AR4	73 (60.3–83.4)	86.3 (76.7–92.9)	0.92
AR5–7	64.1 (58.4–69.4)	95.1 (91.6–97.4)	0.90
AR6	66.2 (59.7–72.3)	93.3 (89.7–95.9)	0.92
AR8	71 (58.8–81.3)	86.3 (76.2–93.2)	0.74
diaphragm	49.7 (42.6–56.9)	97.7 (94.8–99.3)	0.91
small bowel (AR9–12)	45.5 (35.4–55.8)	94.9 (92.2–96.9)	0.80
colon	30.5 (23.2–38.5)	95.8 (92.2–98.1)	0.36
mesentery	33.8 (27.2–41)	96.9 (94.9–98.3)	0.66

**Table 9 cancers-16-01467-t009:** Diagnostic accuracy of MRI in different abdominopelvic regions on a per-patient basis (AR: abdominopelvic region; CI: confidence interval; AUC: area under the curve).

ARs	Pooled Sensitivity (%CI)	Pooled Specificity (%CI)	AUC (SE)
AR0	64.7 (55.9–72.7)	67.2 (61.2–72.7)	0.82
diaphragm	67.3 (57.3–76.3)	66.5 (61.2–71.5)	0.66
mesentery	59.2 (48.8–69)	75.7 (71.3–79.7)	0.90

**Table 10 cancers-16-01467-t010:** Diagnostic accuracy of FDG PET-CT in different abdominopelvic regions on a per-patient basis (AR: abdominopelvic region; CI: confidence interval; AUC: area under the curve).

ARs	Pooled Sensitivity (%CI)	Pooled Specificity (%CI)	AUC (SE)
AR0	92.9 (86.5–96.9)	85.2 (73.8–93)	0.95
AR1	73.5 (64.5–81.2)	92.1 (85–96.5)	0.83
AR3	70.4 (58.4–80.7)	86.9 (77.8–93.3)	0.78
AR5–7	91.5 (85–95.9)	87.5 (74.8–95.3)	0.94
mesentery	45.5 (30.4–61.2)	98.9 (94–100)	0.9

**Table 11 cancers-16-01467-t011:** Diagnostic accuracy of MDCT in different abdominopelvic regions on a per-region basis (AR: abdominopelvic region; CI: confidence interval; AUC: area under the curve).

ARs	Pooled Sensitivity (%CI)	Pooled Specificity (%CI)	AUC
AR0	60.7 (49.7–70.9)	77 (66.8–85.4)	0.72
AR5–7	46.6 (35.9–57.5)	88.7 (81.4–93.8)	0.78
AR6	24.4 (12.4–40.3)	96.4 (89.9–99.3)	0.99
diaphragm	40.7 (28.1–54.3)	86 (73.3–94.2)	0.89
mesentery	43.4 (29.8–57.7)	90.7 (83.6–95.5)	0.68

**Table 12 cancers-16-01467-t012:** Diagnostic accuracy of FDG PET-CT in different abdominopelvic regions on a per-region basis (AR: abdominopelvic region; CI: confidence interval; AUC: area under the curve).

ARs	Pooled Sensitivity (%CI)	Pooled Specificity (%CI)	AUC
AR0	83.7 (74.8–90.4)	89.3 (78.1–96)	0.90
AR1	66.7 (54.8–77.1)	81.8 (69.1–90.9)	0.99
AR3	74.5 (59.7–86.1)	82.5 (70.1–91.3)	0.88
AR4	75.7 (64–85.2)	80.5 (65.1–91.2)	0.81
AR5–7	57.1 (47.4–66.5)	91.8 (81.9–97.3)	0.74
AR8	77.1 (65.6–86.3)	82.5 (67.2–92.7)	0.86

## Data Availability

The data presented in this study are available in this article.

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
