# Peer review of "Imaging of Peritoneal Metastases in Ovarian Cancer Using MDCT, MRI, and FDG PET/CT: A Systematic Review and Meta-Analysis"

_cancers, 2024, doi:10.3390/cancers16081467_

Round 1

Reviewer 1 Report

Comments and Suggestions for Authors

This fantastic effort commendably addresses a prominent issue in the surgical management of advanced ovarian cancer, i.e the role of imaging accuracy in determining disease resectability more than when compared to detection of disease metastases or recurrence. However, the abundance of information presented may overwhelm the reader. In addition, certain points left me perplexed or with slight concerns;

1) (major); the presentation of statistical analyses and presenting methods. The authors need to further elaborate in the methodology section. The outcomes are actually the " Summary estimates of sensitivity and specificity". The meta-analysis should actually be a diagnostic test accuracy (DTA) meta-analysis with 95 % Confidence Intervals (CIs) by use of hierarchical statistical models. A diagnostic odds ratio (DOR) should be calculated to represent the ratio of the odds of the test being positive if the subject had a disease relative to the odds of the test being positive if the subject did not have the disease (if histology was used as a gold standard for comparison). Please argue. Was both fixed effects model and random effects models were used to examine Se and Sp etc? Crosshair plots should be used to display the results of individual studies so for example, should be plotted as a single sensitivity-specificity point along with corresponding CIs to provide a sense of the scatter of study results (and in this way to address study heterogeneity; couple of examples). Also the ROCs should actually be summary receiver operating curves (this needs be corrected in Figure 5). The authors need to mention about I2 statistics on how to estimate study heterogeneity and the threshold of 30% to indicate low heterogeneity. Such guidance exists in the Cochrane Handbook for Systematic Reviews in DTA.

2) Introduction: PDS followed by intraperitoneal chemotherapy is not the standard of care for advanced ovarian cancer. 

3) Does the omission of the words "tubal" and "primary peritoneal" affect the PICO search strategy?

4)Is MDCT the standard of imaging examination worldwide? I am still under the impression for economic reasons that the conventional spiral CT is the standard modality. My notion is supported by your conclusions in p360-363 suggesting MDCT can be used as an alternative. In fact, all three modalities are second-line. It yet remains very difficult to make the argument for routine use of MRI (for detection of mets) or FDG-PET, doesn't it?

5) why do you think the MDCT detection rates were higher on these AR for example (p394-397)?

6) Although it is obvious to me, the authors need to explain why they used a per-patient and also a per-lesion meta-analysis. In an era of stringent economics, is it sensible to support or promote the use of a per-lesion imaging modality?

6) (Discussion) It is interesting to observe the study's design which aimed to assess imaging diagnostic accuracy based on the ARs reported on the PCI score. While the PCI score remains the gold standard due to its widest external validation (and a reference about radiological PCI to aid surgical planning should be made here ), it is important to acknowledge that the challenge persists to ensure consistency in imaging reporting by Radiologists. For instance, a simple synoptic report might only take 20 minutes, yet it may not consistently follow the same pattern (addressing PCI regions). In light of this, it's worth considering the newly proposed synoptic report approach, where predefined templates serve as reminders for radiologists to address specific areas, particularly those posing challenges for resection or are deemed unresectable. Such a synoptic report, which documents the occurrence of disease in 45 anatomical sites relevant to ovarian cancer distribution, has been shown to improve the completeness of pre-treatment CT reporting but adds 30 minutes to the report turnaround time (Andrieu PC et al, 2023). Additionally, incorporating all abdominopelvic regions from the ESGO structured surgical operative template could potentially offer more utility than PCI (PMID: 38001646). This approach might be more ovarian cancer-specific and could better test the efficacy of radiological assessment.

Reviewer 2 Report

Comments and Suggestions for Authors

Here are three suggestions to elevate the level of Paper:

1. Provide a more comprehensive literature review: Expand the review to include recent studies and explore different imaging techniques for peritoneal metastases in ovarian cancer.

2. Enhance the methodology: Provide more details on the study design, patient selection criteria, and imaging protocols used in MDCT, MRI, and FDG PET/CT.

3. Improve the discussion section: Analyze the findings in a more critical and comparative manner, discussing the strengths and limitations of each imaging modality and suggesting future research directions.

I have read and understood Paper carefully. Based on my advice, I will now provide a revised version of the introduction:

Introduction:

Ovarian cancer (OC) is a leading cause of death among gynecologic malignancies, with peritoneal metastases (PMs) representing the most common pathway for the spread of the disease. Accurate detection and mapping of PMs are crucial for planning appropriate therapeutic strategies and predicting the success of cytoreduction surgery. Multidetector CT (MDCT), MRI (including diffusion-weighted imaging), and FDG PET/CT are commonly used imaging modalities for assessing the extent of peritoneal carcinomatosis in OC. This systematic review and meta-analysis aim to update the role of MDCT, MRI, and FDG PET/CT in detecting peritoneal metastases in ovarian cancer by evaluating the existing literature.

Here is a revised version of the discussion section that includes the disadvantage mentioned earlier:

Discussion:

The findings of this systematic review and meta-analysis indicate that MRI and FDG PET/CT exhibit higher sensitivity compared to MDCT on a per-patient basis. However, it is important to note that the limited patient inclusion criteria, focusing only on ovarian/fallopian/primary peritoneal cancer, may restrict the generalizability of these results. Including a broader range of ovarian cancer types and primary peritoneal cancer in future studies would provide a more comprehensive understanding of the diagnostic performance of these imaging modalities.

Additionally, the discussion should address the potential limitations of the included studies, such as variations in imaging protocols, reader experience, and the reference standard used for comparison. Standardization of imaging techniques and consensus on the interpretation criteria for peritoneal metastases across different centers would facilitate more accurate and reliable assessments.

Furthermore, future research should focus on evaluating the cost-effectiveness of MDCT, MRI, and FDG PET/CT in detecting peritoneal metastases and their impact on treatment decision-making. Comparative studies with larger patient cohorts and a wider range of malignancies would enhance the robustness and applicability of the findings.

In summary, while MRI and FDG PET/CT are preferred imaging modalities for detecting peritoneal metastases in OC, the limitations of the included studies and the need for standardization should be considered. Further research addressing these limitations and exploring cost-effectiveness would contribute to the improvement of clinical practice in the management of ovarian cancer.

Round 2

Reviewer 1 Report

Comments and Suggestions for Authors

Thank you for your response; 

Minor issues

1) Figures 5 and 8 are forest plots than crosshair plots. Crosshair plots visualise individual study outcomes representing the sensitivity and false positive rate (FPR = 1- specificity) for each study in the same graph.

2) Figure 6 legend: these are Summary receiver operator curves (SROC) and not ROCs

3)Ultimately, the proposition here inclusive of limitations is that FDG-PET and MRI may be superior than MDCTs for patient identification of PMs; quite revolutionary as a statement. Should this be reflected on the title?

4)The reason for limited result generalisabilty is study heterogeneity and not restricted inclusion criteria as factually, ovarian tubal and primary peritoneal cancer (one entity) have very low incidence anyway. Study heterogeneity could be even visualised with a funnel plot.
